# Population Dynamics between *Erwinia amylovora*, *Pantoea agglomerans* and Bacteriophages: Exploiting Synergy and Competition to Improve Phage Cocktail Efficacy

**DOI:** 10.3390/microorganisms8091449

**Published:** 2020-09-22

**Authors:** Steven Gayder, Michael Parcey, Darlene Nesbitt, Alan J. Castle, Antonet M. Svircev

**Affiliations:** 1Centre for Biotechnology, Brock University, St. Catharines, ON L2S 3A1, Canada; sg10yl@brocku.ca (S.G.); mp17ll@brocku.ca (M.P.); 2Agriculture and Agri-Food Canada, Vineland Station, ON L0R 2E0, Canada; darlene.nesbitt@canada.ca; 3Department of Biological Sciences, Brock University, St. Catharines, ON L2S 3A1, Canada; acastle@brocku.ca

**Keywords:** fire blight, phage carrier, bacteriophages, qPCR, phage–host dynamics, phage cocktails, phage therapy

## Abstract

Bacteriophages are viruses capable of recognizing with high specificity, propagating inside of, and destroying their bacterial hosts. The phage lytic life cycle makes phages attractive as tools to selectively kill pathogenic bacteria with minimal impact on the surrounding microbiome. To effectively harness the potential of phages in therapy, it is critical to understand the phage–host dynamics and how these interactions can change in complex populations. Our model examined the interactions between the plant pathogen *Erwinia amylovora*, the antagonistic epiphyte *Pantoea agglomerans*, and the bacteriophages that infect and kill both species. *P. agglomerans* strains are used as a phage carrier; their role is to deliver and propagate the bacteriophages on the plant surface prior to the arrival of the pathogen. Using liquid cultures, the populations of the pathogen, carrier, and phages were tracked over time with quantitative real-time PCR. The jumbo *Myoviridae* phage ϕEa35-70 synergized with both the *Myoviridae* ϕEa21-4 and *Podoviridae* ϕEa46-1-A1 and was most effective in combination at reducing *E. amylovora* growth over 24 h. Phage ϕEa35-70, however, also reduced the growth of *P. agglomerans*. Phage cocktails of ϕEa21-4, ϕEa46-1-A1, and ϕEa35-70 at multiplicities of infections (MOIs) of 10, 1, and 0.01, respectively, no longer inhibited growth of *P. agglomerans*. When this cocktail was grown with *P. agglomerans* for 8 h prior to pathogen introduction, pathogen growth was reduced by over four log units over 24 h. These findings present a novel approach to study complex phage–host dynamics that can be exploited to create more effective phage-based therapies.

## 1. Introduction

Bacteriophages, or phages, are bacterial viruses that infect and replicate within their host [1]. Lytic phages subsequently rupture and kill their bacterial host, releasing new phage progeny that can infect other nearby hosts to continue this life cycle. Phages can identify their host with great specificity and therefore play a large role in shaping microbial communities [1]. The overuse of antibiotics has led to a global health crisis of widespread antibiotic resistance and has significantly impacted the environment via the soil and its associated microbial communities [2,3]. Phages have therefore seen a recent resurgence of interest for their potential use as antimicrobial agents to replace or supplement antibiotics for the control of human, animal, and plant pathogens [3,4,5]. Understanding the factors that determine the success or failure of phage therapy is important for the development of successful therapeutic applications [4]. Phage–host interactions such as host range, burst size, adsorption rate, and time to lysis are often critical determinants in choosing phages for cocktails or mixtures, while mathematical models have sought to understand how these factors interact to achieve successful control [4,6].

The plant pathogen *Erwinia amylovora* causes the disease commonly referred to as fire blight in apples, pears, and other members of the Rosaceae. Originally native to North America, *E. amylovora* has spread worldwide and is present in many fruit-growing regions of the world [7]. Infection of trees commences when the nutrient-rich stigma of open blossoms is exposed to *E. amylovora* and the pathogen begins to replicate. The pathogen continues to colonize the stigma until high humidity or a wetting event washes cells into the nectarthodes located in the floral cup [8]. This allows the bacteria to enter the tree and spread throughout the vasculature of the host plant, leading to wilt, necrosis, and potential death of the entire tree [9]. Temporal factors and weather conditions can make effective control with biological control agents or biologicals very challenging. Open blossoms can be rapidly colonized by upwards of 30 bacterial phyla of epiphytic microbes [10,11]. From the initial blossom opening to petal fall, this microbial diversity is greatly reduced as *Proteobacteria* come to make up >99% of the microbiota, most of which is also almost entirely *Enterobacteriaceae* and *Pseudomonadaceae* [10]. Management of fire blight largely consists of alteration of the microbiota resulting in pathogen reduction on the open blossoms [10]. Antibiotics are the preferred means of control for growers, but as some countries ban their use and antimicrobial resistance spreads there is a growing interest in the use of biologicals within integrated pest management (IPM) strategies [5,12].

*Pantoea agglomerans* is an epiphytic bacterium found in the orchard ecosystem or phyllosphere. *P. agglomerans* has shown antagonistic properties against *E. amylovora* through competitive exclusion and production of antimicrobials [13]. *P. agglomerans* strain E325 is the active ingredient of the commercial biological product Bloomtime [14]. Additionally, *Erwinia* lytic phages are capable of infecting *P. agglomerans*, and together have been studied for their potential as an additional means of *E. amylovora* control [15]. In this phage-mediated biological control system, phage infected *P. agglomerans* strains (termed the carrier) are applied to the blossom. These carrier cells are used to increase the phage population in vivo and any non-infected cells may act as a biological control agent [15]. Therefore, understanding the dynamics and interactions between *P. agglomerans* antagonism and phage lysis is critical to harness any potential synergy for *E. amylovora* control.

The lytic phages of *E. amylovora* in our collection belong to four species: the *Kolesnikvirus Erwinia virus Ea214* and *Agricanvirus Erwinia virus Ea35-70* of the *Myoviridae* family, and the *Eracentumvirus Erwinia virus Era103* and *Johnsonvirus Erwinia virus Ea9-2* of the *Podoviridae* family. The biggest known determinant of host range is the exopolysaccharide (EPS) production, particularly amylovoran [16,17]. Erwinia virus Era103 phages require amylovoran for infection and prefer hosts with higher EPS production while Erwinia virus Ea214 phages prefer lower producers of EPS [16,17]. The effect of EPS preferences on phage production is greater in strains from the Western regions of North America, but the cause of this is not known [16]. The Erwinia virus Ea35-70 phage does not follow these trends and is generally much less successful at infecting *E. amylovora*, and therefore even less is known about its infection strategy [16].

Studies of in vitro phage–host interactions in liquid culture commonly use optical density to compare bacterial growth dynamics in response to different phage treatments [18,19,20,21,22,23]. Optical density measurements are quick and amenable to high-throughput comparison using 96-well incubating spectrophotometers, and hosts can even be fluorescently labeled to allow differentiation [24]. However, phage quantification in these cultures, especially those with phage mixtures, is frequently avoided due to the extra effort required and inability to quantitatively differentiate multiple phages. Real-time quantitative PCR (qPCR) is a rapid and reproducible alternative to the double agar overlay for phage quantification [25]. qPCR makes it possible, by differentially quantifying the genomes in solution, to individually track multiple bacteria and phages in mixed cultures. Previously, we used qPCR to determine the host range of 10 phages against a global collection of *E. amylovora* [16] and to create a molecular profile of phage infection to measure adsorption, burst size, latency period, and time to burst [26].

The objective of this study was to investigate the complex dynamics between the pathogen *E. amylovora*, the epiphytic phage carrier *P. agglomerans*, and several combinations of *Erwinia* spp. phages. The aim was to identify possible interspecies phage interactions that could enhance the control of the pathogen *E. amylovora* or survival of the carrier *P. agglomerans*. Growth of combinations of different species of *Myoviridae* and *Podoviridae*, and of *E. amylovora* and *P. agglomerans* were assessed by qPCR. This work represents the first step in understanding the complex interactions between phage, pathogen, and epiphyte in the orchard.

## 2. Materials and Methods

### 2.1. Bacterial Isolates

All bacterial strains used in this study are listed in Table 1. All cultures were stored at −80 °C in Microbank cryobeads (Pro-Bank Diagnostics, Richmond Hill, ON, Canada). Cultures were initially spread onto 2.3% Difco nutrient agar (NA; BD, Sparks, MD, USA) using one cryobead. These cultures were grown overnight at 27 °C and stored at 4 °C for 1 to 2 weeks. Working subcultures were obtained from the initial cultures, plated onto NA, and grown at 27 °C overnight or over 2 to 3 days at room temperature.

### 2.2. Bacteriophage Propagation

A bacterial suspension of the phage’s specific propagation host (Table 2) was created by scraping cells from a NA plate into 3 mL of 0.8% nutrient broth (NB; BD, Sparks, MD, USA) and adjusting to an OD_600_ of 0.6. Using a 250 mL baffled Erlenmeyer flask, we added 100 µL bacterial suspension to 75 mL NB, which was incubated at 27 °C and 125 rpm for 3–4 h. To this culture, we added 100 µL of phage stock, and the mixture was incubated overnight. The following morning, we added 1 mL chloroform to the flask, which was shaken at 125 rpm for about 5 min. The culture was centrifuged at 8500× *g* at 4 °C for 15 min, and the supernatant was filtered through a 0.22 µm Steriflip filter (Millipore, Billerica, MA, USA). These working phage stocks were stored at 4 °C in dark amber glass vials.

Titers of working phage stocks were determined with qPCR (Section 2.3) after treatment with DNase I to remove non-encapsidated genomes. To 10 µL phage lysate, we added a mix of 1 µL 2000 U/mL DNase I (M0303S, NEB, Ipswich, MA, USA), 2 µL 10X DNase buffer (B0303S, NEB, Ipswich, MA, USA), and 7 µL water. This was incubated for 40 min at 37 °C and then 20 min at 80 °C. Phage stocks for the host range experiments (Section 2.5) were not treated with DNase prior to qPCR. In all subsequent experiments, we serially diluted phages in NB such that the addition of 100 µL of diluted phage would achieve the intended multiplicity of infection (MOI), or number of phages per host cell.

### 2.3. Quantitative Real-Time PCR (qPCR)

Genome quantification with qPCR was performed as per the study in [16]. Briefly, each 20 µL reaction consisted of 4 µL 5X EVOlution Probe qPCR Mix (Montreal Biotech Inc., Montreal, QC, Canada), 200 nM of each primer, 100 mM probe, and 2 µL template. Primer and probe sequences are given in Appendix A. Reaction conditions began with 10 min at 95 °C, followed by 40 cycles of 10 s at 95 °C and 45 s at 54 °C. The plasmid pTotalStdA [16] contains amplicons for *E. amylovora* and *P. agglomerans*, and the phages ϕEa21-4, ϕEa46-1-A1, ϕEa9-2, and ϕEa35-70. Dilutions of pTotalStdA at 10^11^, 10^8^, and 10^5^ copies/mL were included in every run to generate the standard curve. Raw population quantities (genomes/mL) were calculated from their Ct values relative to the Ct values and copies/mL of the pTotalStdA standard curve. Populations of bacteria and phage were directly interpreted from this curve on the assumption that each cell or phage capsid contained a single genome. qPCR quantification related to the host range (Section 2.5) was performed with a Stratagene Mx3005P qPCR System (Agilent Technologies, Santa Clara, CA, USA) and all others were performed with a qTower^3^ qPCR System (Analytik Jena, Jena, Germany).

### 2.4. Population Dynamics of E. amylovora, P. agglomerans, and Different Phage Cocktails

Initial liquid bacterial cultures were prepared to an OD_600_ of approximately 0.1, or 10^8^ CFU/mL, in 10 mL of NB. NA working subcultures were scraped into 3 mL NB to an OD_600_ of 0.6, or 10^9^ CFU/mL. In a 50 mL conical tube, we added 100 µL cell suspension to 10 mL refrigerated NB. In order to produce fresh and exponentially growing cells for the early morning, we incubated these cultures in a programmable shaking incubator (New Brunswick Innova 44R, Eppendorf, Mississauga, ON, Canada), which was maintained at 4 °C at 0 rpm most of the night and switched to 27 °C at 175 rpm in the early morning to give roughly 5 h growth. The OD_600_ was measured until 0.1 was achieved, at which time these cultures were stored on ice until they were used to prepare the experimental cultures.

Experimental cultures throughout Section 3.1 and Section 3.3 contain different bacterial and phage populations but were prepared in a modular fashion following a consistent methodology. Hosts tested include individual or combined cultures of Ea17-1-1, EaD7, 20060013, 20070126, Pa31-4, and Pa39-7. PaMix, EaMix, and EaWest were equal combinations of Pa31-4 and Pa39-7, Ea17-1-1 and EaD7, and 20060013 and 20070126, respectively. Phages tested include ϕEa21-4, ϕEa46-1-A1, and ϕEa35-70 individually and in combination. The bacteria and phages tested in each experiment are indicated within each figure. Into a 50 mL conical tube, we pre-aliquoted 9.5–9.9 mL NB such that the final culture volume was 10 mL after 100 µL of all host cultures and phages were each added individually. First, phage dilutions were added such that the intended MOI would be achieved. Except where indicated within the figure, the intended MOI for all phages was 1. Then, 100 µL of the bacterial cultures were added such that the cultures initially contained 10^6^ CFU/mL total *P. agglomerans* or 10^6^ CFU/mL total *E. amylovora*. Cultures were grown at 27 °C and 175 rpm for 24 h. Samples of 100 µL were taken at 0, 2, 4, 6, 8, and 24 h into microcentrifuge tubes; heated at 80 °C for 20 min; then frozen until qPCR analysis. DNA isolation and DNase treatment were not performed prior to qPCR. Samples were thawed before qPCR, and 2 µL was used as template following the qPCR methodology in Section 2.3. All tests were performed with three experimental replicates, each measured with qPCR once.

### 2.5. Phage Host Range Assay

Phage host range was studied as per the work in [16], with the exception that phage were initially added at a starting concentration of 10^6^ genomes/mL. Hosts tested included all strains of *P. agglomerans*, *Pantoea eucalypti*, *Erwinia gerundensis*, *Rahnella aquatilis*, and *Pantoea* sp. in Table 1 with all 10 phages listed in Table 2. Host cultures were scraped from working subculture NA plates into 3 mL NB to an OD_600_ of 0.6. In a 2 mL microcentrifuge tube, we added 100 µL host cells and 100 µL phage to 800 µL NB. These cultures were incubated at 27 °C and 200 rpm for 8 h. Cultures were heated for approximately 10 min over 70 °C to lyse the cells, stopping growth and releasing genomic DNA, and were then frozen until qPCR analysis. DNA isolation and DNase treatment were not performed prior to qPCR. Samples were thawed before qPCR, and 2 µL was used as a template following the qPCR methodology in Section 2.3. All combinations of host and phage were performed with three experimental replicates, each measured with qPCR once.

### 2.6. Phage and Carrier Combinations for E. amylovora Reduction

Experimental cultures were prepared the same as per Section 2.4, with the exception that EaMix was added to the culture either immediately (0 h) or after 2 or 8 h of culture growth. Cultures were grown at 27 °C and 175 rpm. Samples of 100 µL were taken upon addition of EaMix and 24 h after EaMix was added. These samples were heated at 80 °C for 20 min, then frozen until qPCR analysis. DNA isolation and DNase treatment were not performed prior to qPCR. Samples were thawed before qPCR, and 2 µL was used as template following the qPCR methodology in Section 2.3. All experiments were performed with three experimental replicates, each measured with qPCR once.

### 2.7. Genomic Sequencing of P. agglomerans and other Epiphytes

Genomic DNA was isolated from overnight bacterial cultures using the Bacterial Genomic DNA Isolation Kit (17900; Norgen Biotek Corp., Thorold, ON, Canada). Paired-end library prep and MiSeq Illumina sequencing were performed by Genome Québec. Low-quality paired end reads were trimmed using Sickle, version 1.33 [32,33]. Sequences were assembled de novo using SPAdes assembler, version 3.11.1, in careful mode. The genomes were uploaded to GenBank and species identity was confirmed upon submission.

### 2.8. Data Analysis

Data analysis was performed in R version 3.5.1 [34]. Data manipulation and calculations were performed using the dplyr package, version 0.7.6 [35]. Genome quantities were log_10_ transformed before calculating mean population titers and bacterial reductions. Bacterial reductions at each time point were calculated as the difference between the log_10_ quantities of the treated sample and the untreated control. MOI_g_, a genomic interpretation of MOI in real time, was calculated by dividing the untransformed genome quantity of phage by that of bacteria. In mixed cultures, the MOI_g_ can be determined for each phage individually or as a sum of all phage genomes and for both bacterial host species separately. Linear regression was performed between the log_10_ OD_600_ and log_10_ total host genome quantity of measured 24 h samples. The linear equation was then used to calculate the predicted genomic titer from the sample OD_600_ values and was compared to the qPCR measured quantities of *E. amylovora* in the presence and absence of *P. agglomerans*. Figures were generated with ggplot2, version 3.0.0 [36], using the packages scales, version 1.0.0 [37], and ggthemes, version 4.0.1 [38]. Statistical comparisons were performed using ANOVA followed by post hoc multiple comparison with Tukey’s test, and differences were considered significant when *p* < 0.05.

## 3. Results

### 3.1. Population Dynamics of E. amylovora and Different Phage Cocktails

The growth of both bacteria and phage can be monitored with qPCR as an increase in their respective genomes. Populations of different strains of *E. amylovora* infected with different phage cocktails were measured over time. In liquid culture at 27 °C, the growth of all uninfected *E. amylovora* strains were similar, reaching 10^9^ genomes/mL after 8 h and not exceeding 10^10^ genomes/mL after 24 h (Figure 1). Different strains were variably affected by the composition of the phage cocktails. The log_10_ reduction of each culture compared to the control was also determined (Appendix A). Reduction of Ea17-1-1 began after 8 h but only when ϕEa21-4 was present in the cocktail (●,●,●,●; *p* < 0.05). The reduction of EaD7 began within 4 h in all cultures where ϕEa46-1-A1 was present (●,●,●,●; *p* < 0.05). Additionally, no culture’s growth was affected by the presence of ϕEa35-70 alone (●; *p* > 0.05). However, significant reduction after 24 h of Ea17-1-1 and EaD7 was only maintained when ϕEa35-70 was combined with ϕEa21-4 (●,●; *p* < 0.05) and ϕEa46-1-A1 (●,●; *p* < 0.05), respectively. In a mixture of the two strains (EaMix), no single phage effectively reduced growth, and only cocktails containing both ϕEa21-4 and ϕEa46-1-A1 started to slow growth after 6 h (●,●; *p* < 0.05). This remained consistent until 24 h, and here additional synergy was observed with the inclusion of ϕEa35-70 (●; *p* < 0.05).

In liquid cultures, the phage populations were tracked alongside the bacterial hosts. The phage ϕEa21-4 was able to grow to over 1 × 10^10^ genomes/mL with Ea17-1-1 and EaMix in 8 h and maintain this titer until 24 h (Figure 1). The growth of ϕEa21-4 was also not influenced by the presence of other phages if Ea17-1-1 was also present. ϕEa21-4 grew slower on EaD7 and did not reach a growth plateau by 8 h. The initial growth of ϕEa21-4 was reduced in the presence of ϕEa46-1-A1 (●,●), and by 8 h was 2.41 log units less than ϕEa21-4 alone (●). Whereas ϕEa46-1-A1 reduced ϕEa21-4 growth only up to 8 h, both ϕEa46-1-A1 and ϕEa35-70 together reduced ϕEa21-4 growth until 24 h, at which time it only reached a titer of 1 × 10^7^ genomes/mL (●). Notably, the growth of ϕEa21-4 mirrored the growth of EaD7, whereby a reduction of EaD7 resulted in a reduction of ϕEa21-4 (Figure 1). ϕEa46-1-A1, with its preferred host EaD7, increased nearly three log units in the first 2 h, the fastest increase seen in any of the phage–host combinations (Figure 1). On Ea17-1-1, the phage took 4 h to increase the same amount, and on EaMix, production was similar to that of EaD7 and increased by 2.5 log units in the first 2 h. ϕEa35-70 grew similarly in all cultures, reaching around 10^9^–10^10^ genomes/mL at 8 h with reductions in cultures where host cell numbers were also reduced.

With populations of both host and phages measured simultaneously using qPCR, we inferred the changing MOI as the ratio of phage genomes per host genome (MOI_g_). The control of *E. amylovora* (Appendix A) and the MOI_g_ (Appendix A) are potentially linked. The time required to reduce the *E. amylovora* population by 0.25 log units was phage–host-dependent: phage ϕEa21-4 with Ea17-1-1 and ϕEa46-1-A1 with EaD7 taking 6 and 2 h, respectively. These times coincided with the time at which the MOI_g_ approached 100 in each reaction. This was phage–host-dependent where only ϕEa21-4 and ϕEa46-1-A1 reached this level on Ea17-1-1 and EaD7, respectively. With EaMix, the MOI_g_ of ϕEa46-1-A1 only increased to 22 in 2 h, while ϕEa21-4 continued to reach an MOI_g_ of 100 in 6 h (Appendix A). Reduction of the *E. amylovora* population in EaMix only exceeded 0.25 log units at 4 h in cocktails containing ϕEa46-1-A1 (●,●,●,●). Of these cocktails, only the two with ϕEa21-4 (●,●) continued to further reduce the *E. amylovora* population to 6 and 8 h and only the three-phage cocktail (●) continued to control EaMix until 24 h. By 24 h, the three-phage cocktail (●) reduced the *E. amylovora* population by an additional log unit further than the ϕEa21-4 + ϕEa46-1-A1 cocktail (●).

Next, we investigated whether higher-starting MOIs would inhibit the growth of EaMix earlier (Figure 2). Moreover, we investigated whether this phage combination would still be effective against a mix of *E. amylovora* strains 20060013 and 20070126 from Utah, USA (EaWest), which we previously found to have some resistance to phages ϕEa46-1-A1 and ϕEa21-4, respectively [16]. EaWest was still reduced by this phage combination, but significant control at 24 h required starting MOIs of 10 or 100 (*p* < 0.05). Interestingly, the maximal phage titers and initial rate of replication decreased as the starting MOI increased.

### 3.2. Host Range of Phages on Potential Carriers

To identify which strains of *P. agglomerans* would be effective phage carriers, we performed a host range assay of 30 strains of *P. agglomerans* and other epiphytic bacteria with 10 phages [16]. The *Podoviridae* phages ϕEa10-7, ϕEa31-3, and ϕEa46-1-A2 were the only phages that did not increase by at least two log units (>10^8^ genomes/mL) on any hosts tested (Figure 3). Only two hosts (Pa17-1 and Pa39-7) were able to propagate at least one phage from each species to over 10^8^ genomes/mL.

*Myoviridae* phage ϕEa21-4 was propagated to over 10^9^ genomes/mL in 10 hosts, including all four strains of *Erwinia gerundensis* and six strains of *P. agglomerans* (Figure 3). Interestingly, of those six strains of *P. agglomerans,* four were not hosts of *Myoviridae* phage ϕEa35-70. Additionally, of the 26 hosts that propagated phages ϕEa21-4 or ϕEa35-70 to over 10^8^ genomes/mL, only seven hosts enriched both phages. In contrast, all five of the hosts that enriched ϕEa46-1-A1 over 10^8^ genomes/mL were good hosts of the other *Podoviridae* phage ϕEa9-2. We chose to further investigate Pa39-7 as the carrier of the phages and Pa31-4 to grow unaffected by the phages. We hypothesized that in combination these two *P. agglomerans* strains would synergize to maximize control of *E. amylovora*.

### 3.3. Population Dynamics of P. agglomerans and Different Phage Cocktails

The growth of *P. agglomerans* was affected by the composition of phage cocktails (Figure 4, Appendix A). The growth of carrier Pa31-4 was not significantly affected by the presence of any phages (*p* > 0.05), whereas Pa39-7 was reduced by an average of 1.94 log units in cocktails containing ϕEa35-70 by 8 h (●,●,●,●; *p* < 0.05). Unexpectedly, the growth of the two carriers in combination (PaMix) was also affected by the presence of ϕEa35-70, with an average log reduction of 1.64 at 8 h (*p* < 0.05). Instead of Pa31-4 growing while only Pa39-7 was inhibited, which was what we expected, Pa31-4 growth was also reduced in this environment. After 24 h, populations were no longer different (*p* > 0.05).

On carrier Pa31-4, ϕEa21-4 reached nearly 10^10^ genomes/mL by 8 h and maintained this titer until 24 h (●; Figure 4). Moreover, ϕEa21-4 growth was unaffected by the presence of other phages in the cocktail. Both ϕEa46-1-A1 (●) and ϕEa35-70 (●) grew minimally on Pa31-4, and in the presence of ϕEa21-4 (●,●,●) their final titers after 24 h were less than their initial titers. On Pa39-7, ϕEa21-4 (●) growth was perturbed in the presence of ϕEa35-70 (●,●) by 1.27 log units by 8 h, which coincided with a reduction of the Pa39-7. ϕEa35-70 growth (●) was affected by ϕEa21-4 (●,●) starting at 6 h, and also by ϕEa46-1-A1 (●) at 24 h. Growth of phages and hosts were similar for both Pa39-7 and PaMix (Figure 4).

Reduction of *P. agglomerans* populations (Appendix A) again coincided with the ratio of phage genomes to bacterial genomes (Appendix A). On Pa31-4, which was not reduced by any phage, only the MOI_g_ of ϕEa21-4 increased over time but never exceeded 10 throughout (Appendix A). On Pa39-7, the MOI_g_ of ϕEa35-70 surpassed 100 at 4 h, at which point the reduction of Pa39-7 exceeded 0.5 log units. The total MOI_g_ remained over 100 until 8 h, and the reduction continually increased to an average 1.94 log units at 8 h. Notably, on PaMix, the total MOI_g_ only reached 56.9 at 4 h with ϕEa35-70-containing cocktails but the *P. agglomerans* population was still reduced by 0.48 log units. While not as repressed as Pa39-7 alone, PaMix was still reduced by 1.64 log units by 8 h by cocktails containing ϕEa35-70. Despite the high MOI_g_, Pa39-7 and PaMix were able to recover and grow to the control by 24 h (Figure 4).

We then investigated whether we could further perturb ϕEa35-70 enough to allow carrier growth to continue unaffected. We chose MOIs of 10, 1, and 0.01 of ϕEa21-4, ϕEa46-1-A1, and ϕEa35-70, respectively, with PaMix (3ϕ_MOI_) to compare against the MOI of 1 (3ϕ), which was used in the previous experiments (Figure 5). The growth of PaMix with 3ϕ was again reduced from 4-8 h (*p* < 0.05) but with 3ϕ_MOI_ was nearly identical to the uninfected control over 24 h (*p* > 0.05). The phage ϕEa21-4 genomic titer in the 3ϕ_MOI_ cocktail remained approximately one log unit higher than 3ϕ for the first 8 h, but by 24 h the titers of ϕEa21-4 on both had reached 10^11^ genomes/mL. Phage ϕEa46-1-A1 was mostly unaffected by the change in MOI of the other phage, with differences not exceeding 0.5 log units. Unfortunately, ϕEa35-70 in 3ϕ_MOI_ on PaMix did not reach 10^7^ when it reached its maximum after 4 h and slowly declined thereafter. Considering that ϕEa35-70 began at 10^4^ initially, there was still considerable phage production (Figure 5).

### 3.4. Phage and Carrier Combinations for E. amylovora Reduction

To model how this chosen MOI (3ϕ_MOI_) would impact the survival of PaMix and subsequently control pathogen populations, we grew these treatments for different times before *E. amylovora* was introduced. When co-introduced (0 h), PaMix alone reduced EaMix by 0.83 log units over 24 h (Figure 6). Treatments containing phage were more effective than PaMix alone, and the addition of PaMix with phage was no more effective than phage alone. With 2 h treatment growth before EaMix was added, none of the treatments were significantly more effective than they were at 0 h, and only PaMix + 3ϕ_MOI_ was more effective than PaMix alone. After 8 h prior treatment growth, PaMix + 3ϕ became less effective, but not significantly, when compared with PaMix alone, while PaMix + 3ϕ_MOI_ was the most effective treatment overall. Synergy between phage and carrier was seen only with 3ϕ_MOI_ at 2 and 8 h, wherein the treatment was more effective than phage or PaMix alone, reducing EaMix by 4.3 log units at 8 h (Figure 6).

### 3.5. Comparison of qPCR and Spectrophotometry for Quantification of Mixed Bacterial Populations

Measuring the optical density of a culture is a common alternative to directly counting viable colonies, especially when multiple measurements are made over time. To show that measuring host genomes with qPCR is comparable to optical density for bacterial quantification, we measured the OD_600_ of 90 experimental cultures at 24 h at the time they were sampled for qPCR. We plotted these OD_600_ values against the sum of all bacterial genomes detected with qPCR (Figure 7A). A regression of the total genomes determined with qPCR and OD_600_ of cultures showed that these were strongly correlated with an *R*^2^ of 0.93. Because optical density cannot differentiate bacterial species in liquid culture, we compared the qPCR titers of *E. amylovora* to the predicted bacterial genome titers calculated from the OD_600_ using the regression (Figure 7B). In cultures where EaMix was the only bacteria, optical density and qPCR populations were comparable between 10^7^–10^10^ genomes/mL, with an average difference of only 0.026 log units. However, when PaMix was also present, *P. agglomerans* often greatly outgrew *E. amylovora* by 24 h, and optical density populations were an average of 1.97 log units higher than qPCR-determined titers of *E. amylovora*. This showed that not only is qPCR quantification highly comparable to optical density for cell quantification, but it can quantitatively differentiate different hosts in the culture, which is not easily done with optical density.

## 4. Discussion

One of the greatest challenges in creating effective phage therapies is the translation of results from in vitro model systems to in vivo applications [4]. The goal of phage therapy is to modify the local microbiome in a way that prevents the pathogen from establishing within its niche and thereby preventing disease progression. To this end, model systems are first developed to gain a better understanding of the population dynamics within complex natural settings. As metagenomics and molecular technologies become more sophisticated, so too should the models of phage therapy. The aim of this study was to elucidate the dynamics between the pathogen *E. amylovora*, the carrier *P. agglomerans*, and cocktails of three potential therapeutic phages to facilitate the design of an effective phage and carrier combination treatment.

The effective control of a mixed *E. amylovora* population required the use of multiple phages (Figure 1, Appendix A). In previous work, phage ϕEa35-70 showed no potential individually as an effective phage for biocontrol [16]. While ϕEa21-4 and ϕEa46-1-A1 were each only effective against one strain, Ea17-1-1 and EaD7, respectively, ϕEa35-70 synergized with both phages against their respective preferred host. Against a mixture of both strains (EaMix), ϕEa35-70 also provided enhanced control with ϕEa21-4 and ϕEa46-1-A1 together, maintaining the highest control at 24 h. The reason for this synergy is unknown, but further investigation into the large genome and infection strategy of ϕEa35-70 could yield novel insights into phage interactions. We also found this combination of phages to be effective at reducing the population of phage-resistant *E. amylovora* strains (Figure 2). This is essential to the continued effectivity of a phage-mediated biocontrol to prevent the development of phage resistance, given that different geographic regions, such as the west coast of North America, show varying levels of sensitively to these phages [16].

To maximize the biocontrol potential of a phage-carrier system, it is imperative that the carrier can quickly colonize the stigma surface. This is necessary for both competitive exclusion of *E. amylovora* and efficient production of the therapeutic phage. However, while including ϕEa35-70 in the cocktail gave synergistic control of *E. amylovora*, this phage greatly impaired growth of *P. agglomerans*. Because we observed phage ϕEa21-4 slowed replication of ϕEa35-70 without affecting the growth of *P. agglomerans* (Figure 4), we chose the MOIs of 10, 1, and 0.01 for ϕEa21-4, ϕEa46-1-A1, and ϕEa35-70, respectively, to further investigate this dynamic. This optimized cocktail (3ϕ_MOI_) no longer inhibited the planktonic growth of PaMix compared to the three-phage cocktail with equal MOIs of 1 (3ϕ). Unsurprisingly, phage production in PaMix was also affected proportionally by the change in starting MOI (Figure 5). The higher ϕEa21-4 production in 3ϕ_MOI_ is potentially ideal for *E. amylovora* control, but the reduced titers of ϕEa35-70 could lead to decreased synergistic effects.

To emulate how biocontrol would occur in the field, we allowed our *P. agglomerans* and phage treatments to grow for 0, 2, and 8 h prior to addition of the pathogen (Figure 6). When co-introduced with EaMix, PaMix alone was least effective, but became more effective as it had more time to grow. While the addition of 3ϕ to PaMix was more effective than PaMix alone at 0 h, there was no significant advantage over only 3ϕ. Furthermore, any benefit from 3ϕ was lost over time as it became less effective than PaMix alone by 8 h. Conversely, the optimization of the MOI of 3ϕ (3ϕ_MOI_) precluded the growth inhibition of the carrier, allowing the antagonistic effects of both *P. agglomerans* and the phages for enhanced control of the pathogen. As such, PaMix with 3ϕ_MOI_ was the most effective treatment at all time points.

The use of qPCR allowed the simultaneous tracking of pathogen, carrier, and phage populations in the growing cultures. This level of detail granted insights into the complex dynamics and interactions that occurred. While some studies or applications may require quantification of intact and infectious particles in specific environments through plaque assays, measuring the replication of phage genomes is an accurate and efficient way of observing changes in a mixed phage population. We showed previously that our qPCR methodology strongly correlates with dilution plating methods for quantification of both *E. amylovora* and *P. agglomerans* [26]. Here, we show this is also the case with optical density measurements of bacterial cultures (Figure 7A). Additionally, we show that optical density is not able to accurately quantify the minority population in mixed cultures (Figure 7B). While our qPCR methodology can differentiate *E. amylovora* and *P. agglomerans*, it is not specific enough to distinguish strains of the same species. Experimentation using mixed strain cultures is important for several reasons. The amount of capsular exopolysaccharides (EPS) produced by *E. amylovora* strains is known to significantly impact phage preference and infection [16,17]. EaMix consists of Ea17-1-1 and EaD7, which produce low and high amounts of EPS, respectively, and is therefore meant to represent the presence of both extremes found in natural populations. For example, choosing EaD7 exclusively would have resulted in high levels of control within in vitro experiments, but would be poorly representative of natural mixed infections found in vivo.

Pa31-4 and Pa39-7 in PaMix were chosen on the basis of the hypothesis that Pa39-7 would replicate the phages while Pa31-4, being phage resistant, would grow uninhibited and enhance the antagonistic effect against *E. amylovora*. It was therefore unexpected that Pa31-4 would be perturbed when combined with Pa39-7 and exposed to phage cocktails, which included ϕEa35-70 (Figure 4). We observed that ϕEa35-70 was rapidly produced, presumably by Pa39-7, to an MOI_g_ high enough that it inhibited Pa31-4 (Appendix A). This interaction highlights how phages can exhibit unexpected cascading effects on more diverse microbial populations, similar to how Hsu et al. showed phages applied to a gut microbiota induced cascading effects on non-target bacteria, and subsequently on gut metabolites as well.

The use of non-pathogenic strains or related species as phage carriers has been investigated but has generally received very little attention [15,39,40]. The host range of many phages are not as broad as the *Erwinia* phages studied here, and finding another wild-type host may be challenging [41]. Altered phage isolation methods can help find broader host range phages [41], or their host range can be directly modified with targeted mutagenesis [42] or editing phage tail genes [43,44]. Moreover, as more methods are developed for identifying the host genes necessary for phage infection [45], host genomes could also be edited to make them possible phage carriers [46]. Carriers could also be edited to provide additional antagonistic properties, potentially achieving new means of pathogen control [47]. Death of the bacterial carrier could also be a hindrance, but as we showed here this can be avoided or reduced with a greater understanding of the dynamics and interactions involved.

The roles that phages play in microbiomes remains under-investigated, especially in those of plants [48]. In animal hosts, phages have been shown to be linked to pathogenic microbial dysbiosis and can even interact with eukaryotic host cells directly [48,49]. Given the parallels between animal gut and rhizosphere microbiomes, it is suggestive of a larger, unrecognized role played by phages in plant health [48]. Recently, it was shown that phage alone affected the metabolism of *Brassica oleracea* var. *gongylodes* [50]. Given the continually emerging evidence that phages play a significant role in shaping microbial populations, a better understanding of the complex dynamics will be crucial for better implementations of phages in therapeutic applications. We have shown that even in our rudimentary model system there is enormous complexity in the dynamics and interactions that can be observed. The interactions that phages have on each other was of particular significance.

While we aimed to investigate the dynamics of complex populations, only a small fraction of the complexity of the actual microbiome on the blossom surface was the focal point of these studies. It remains unknown as to how these results will translate from liquid culture to the microenvironment of the blossom surface. To follow up with our findings, we aim to study these populations on the blossom surface in blossom assays and field experiments. Of particular importance is whether the 3ϕ_MOI_ cocktail will still allow the carrier to grow adequately in planta to cover the stigma surface and whether it will still synergize with the phages present. Additionally, further studies with more controlled and targeted changes to these cocktail and carrier combinations will allow us to further understand their interactions, allowing us to strive to maximize their potential therapeutic activity.

## 5. Conclusions

Our use of qPCR to study complex populations in liquid cultures made it possible to observe several unexpected dynamics that were exploited to create a phage and carrier combination for the control of *E. amylovora*. The jumbo *Myoviridae* phage ϕEa35-70, while ineffective alone, synergized with both ϕEa21-4 and ϕEa46-1-A1 for increased control of *E. amylovora* over 24 h. However, at equal MOIs, this phage cocktail also reduced the growth of the bacterial carrier and antagonist *P. agglomerans*. To avoid this, we exploited the competition between ϕEa21-4 and ϕEa35-70 and chose to infect *P. agglomerans* with MOIs of 10, 1, and 0.01 for ϕEa21-4, ϕEa46-1-A1, and ϕEa35-70, respectively. This modified phage combination no longer inhibited the growth of *P. agglomerans* and subsequently provided enhanced control of the pathogen. From these data, we observed examples of interspecies phage competition, synergy between multiple phages, and synergy between phages and a bacterial antagonist. All of these interactions could be significant factors to consider when designing phage cocktails or incorporating phages into IPM strategies.

## Figures and Tables

**Figure 1 microorganisms-08-01449-f001:**
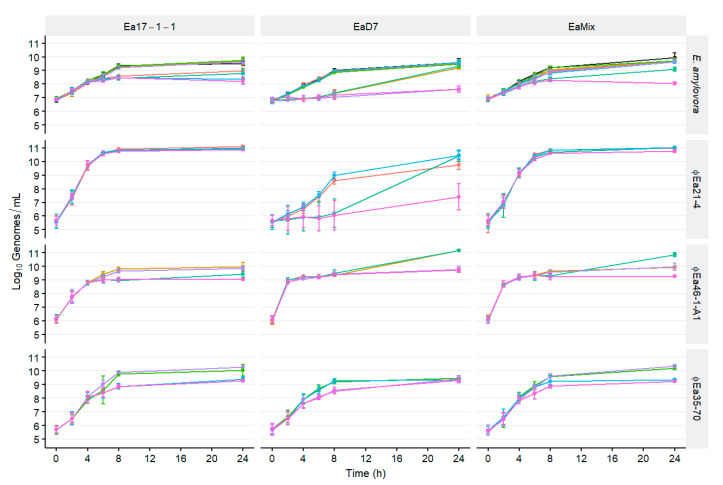
Populations of *E. amylovora* infected with different phage combinations over time. The infected strains Ea17-1-1, EaD7, and an equal mixture of both (EaMix) are indicated in the top banners. The banners on the right indicate the host or phage quantified. Each host was infected by all possible phage combinations, which are indicated by color: ϕEa21-4 (●), ϕEa46-1-A1 (●), ϕEa35-70 (●), ϕEa21-4 + ϕEa46-1-A1 (●), ϕEa21-4 + ϕEa35-70 (●), ϕEa46-1-A1 + ϕEa35-70 (●), ϕEa21-4 + ϕEa46-1-A1 + ϕEa35-70 (●), no phage (●). Genomic titers of *E. amylovora* and each phage were determined with qPCR. Data are the mean ± SD of three replicates.

**Figure 2 microorganisms-08-01449-f002:**
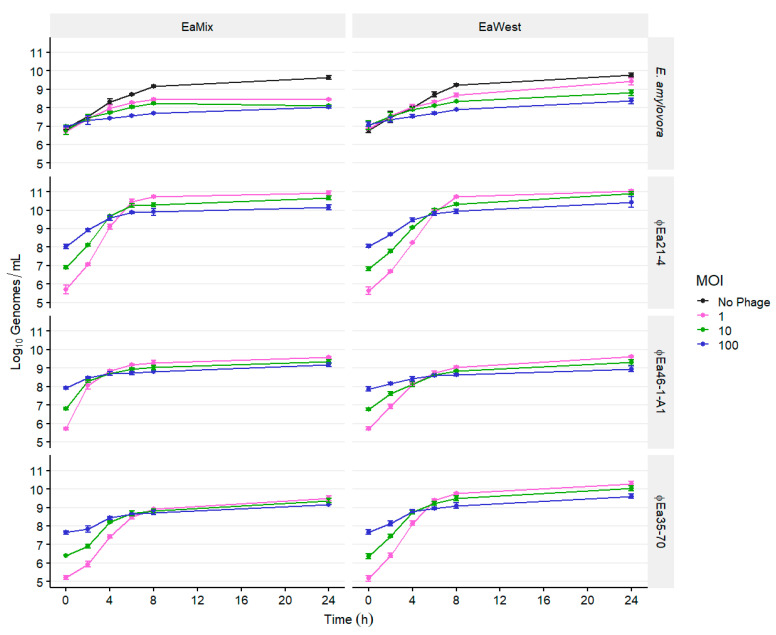
Populations over time of *E. amylovora* cultures infected with the three-phage cocktail at different starting multiplicities of infections (MOIs). The infected cultures were an equal combination of Ea17-1-1 and EaD7 (EaMix), and 20060013 and 20070126 (EaWest), and are indicated in the top banners. The banners on the right indicate the host or phage quantified. Each culture was infected simultaneously by the phages ϕEa21-4, ϕEa46-1-A1, and ϕEa35-70, each at the indicated MOI. Genomic titers of *E. amylovora* and each phage were determined with qPCR. Data are the mean ± SD of three replicates.

**Figure 3 microorganisms-08-01449-f003:**
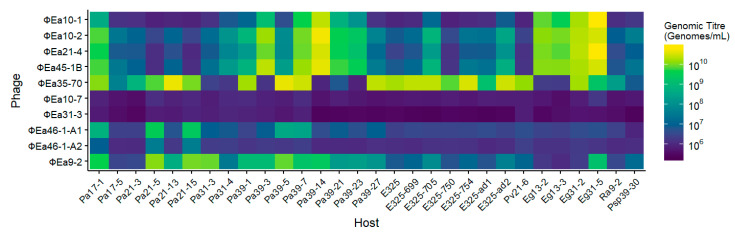
Host range of 10 *E. amylovora* phages against 30 strains of *P. agglomerans* and other epiphytic bacteria. The growth of phage is indicated by color, where yellow shows phage grew to 10^11^ genomes/mL and blue indicates no growth beyond the initial titer of 10^6^ genomes/mL after 8 h incubation. All hosts were initially identified as *P. agglomerans* through detection with PCR primers and subsequently corrected with genomic sequence data. Host species are indicated by their initial lettering: Pa and E325 (*P. agglomerans*); Pv (*P. eucalypti*); Eg (*E. gerundensis*); Ra (*R. aquatilis*); Psp (*Pantoea* sp.). Indicated titers are the geometric mean of three replicates.

**Figure 4 microorganisms-08-01449-f004:**
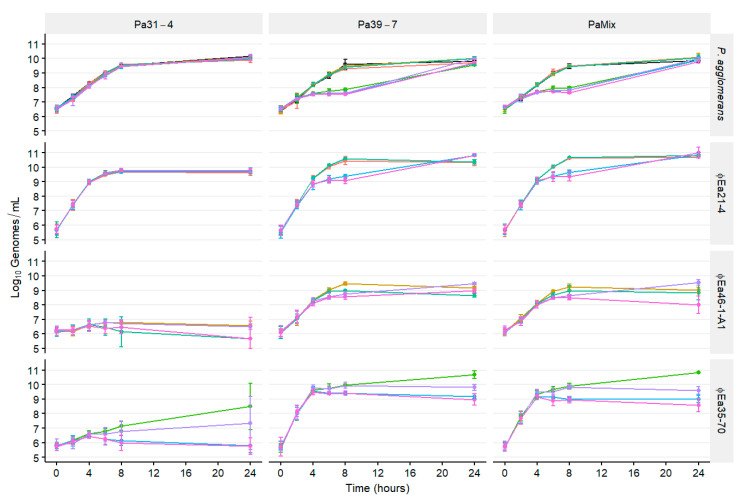
Populations over time of *P. agglomerans* cultures infected with different phage combinations. The infected strains Pa31-4, Pa39-7, and an equal combination of both (PaMix) are indicated in the top banners. The banners on the right indicate the host or phage quantified. Each host was infected by all possible phage combinations, which are indicated by color: ϕEa21-4 (●), ϕEa46-1-A1 (●), ϕEa35-70 (●), ϕEa21-4 + ϕEa46-1-A1 (●), ϕEa21-4 + ϕEa35-70 (●), ϕEa46-1-A1 + ϕEa35-70 (●), ϕEa21-4 + ϕEa46-1-A1 + ϕEa35-70 (●), no phage (●). Genomic titers of *P. agglomerans* and each phage were determined with qPCR. Data are the mean ± SD of three replicates.

**Figure 5 microorganisms-08-01449-f005:**
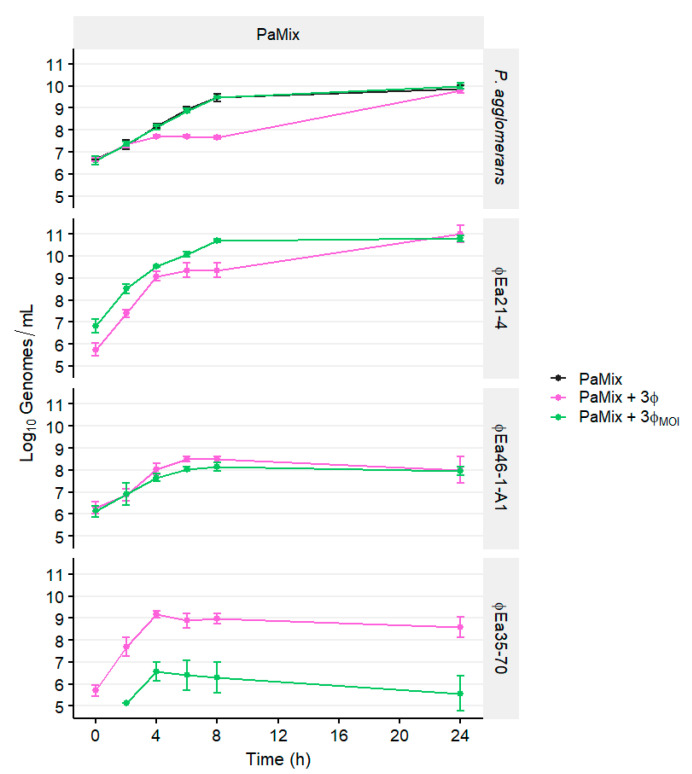
Effect of MOI on the growth of PaMix over 24 h. PaMix was grown with no phage, or infected with the three phages ϕEa21-4, ϕEa46-1-A1, and ϕEa35-70, each at an MOI of 1 (3ϕ) or at 10, 1, and 0.01, respectively (3ϕ_MOI_). The banners on the right indicate the host or phage quantified. Genomic titers of *P. agglomerans* and each phage were determined with qPCR. Data are the mean ± SD of three replicates. The missing data point for ϕEa35-70 at 0 h (●) was below the limit of detection.

**Figure 6 microorganisms-08-01449-f006:**
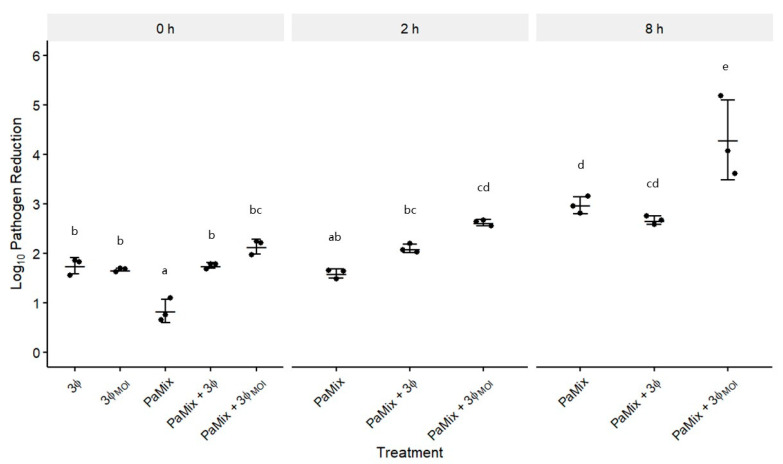
Effect of a designed MOI on the efficacy of PaMix and phage treatment at controlling *E. amylovora*. Treatments were grown for 0 (co-inoculated), 2, or 8 h before EaMix inoculation. The three phages ϕEa21-4, ϕEa46-1-A1, and ϕEa35-70, each at an MOI of 1 (3ϕ) or at 10, 1, and 0.01, respectively, (3ϕ_MOI_) were tested alone at 0 h or with PaMix at each time comparison. *E. amylovora* populations were measured 24 h after inoculation, and reduction was relative to EaMix growth alone. Data are the mean ± SD of three replicates.

**Figure 7 microorganisms-08-01449-f007:**
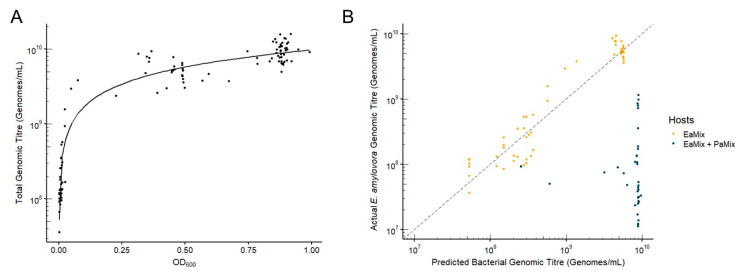
Correlation between OD_600_ and qPCR for quantification of *E. amylovora* in liquid culture. (**A**) A regression of total bacterial genomes (*E. amylovora* + *P. agglomerans*) in solution compared to OD_600_ was used to show the correlation and accuracy of qPCR (*R*^2^ = 0.93). (**B**) The regression was used to calculate the predicted titer of bacterial genomes on the basis of the OD_600_ of the solution, and this was compared to the actual qPCR-determined titer of *E. amylovora* genomes when *E. amylovora* was the only bacterium in solution (●) and when *P. agglomerans* was also present (●). The dotted line represents a 1:1 correlation between the predicted bacterial genomic titer calculated from the OD_600_ and the qPCR titer of *E. amylovora*.

**Table 1 microorganisms-08-01449-t001:** Strains of bacteria used in this study.

Strain	Accession Number	Reference
*Erwinia amylovora*		
Ea17-1-1	JAAEVW000000000	[16]
EaD7	JAAEUT000000000	[16]
Ea6-4	JAAEVD000000000	[16]
Ea29-7	JAAEVM000000000	[16]
20060013	JAAEXH000000000	[16,27]
20070126	JAAEXF000000000	[16,27]
*Pantoea agglomerans*		
Pa17-1	JACSXN000000000	[28]
Pa17-5	JACSXM000000000	[28]
Pa21-3	JACSXL000000000	[28]
Pa21-5	JACSXK000000000	[28]
Pa21-13	JACSXI000000000	[28]
Pa21-15	JACSXH000000000	[28]
Pa31-3	JACSXF000000000	[28]
Pa31-4	JACSXE000000000	[28]
Pa39-1	JACSXC000000000	[28]
Pa39-3	JACSXB000000000	[28]
Pa39-5	JACSXA000000000	[28]
Pa39-7	JACSWZ000000000	[28]
Pa39-14	JACSWY000000000	[28]
Pa39-21	JACSWX000000000	[28]
Pa39-23	JACSWW000000000	[28]
Pa39-27	JACSWV000000000	[28]
E325	JACSWT000000000	[14]
E325-699	JACSWS000000000	[14]
E325-705	JACSWR000000000	[14]
E325-750	JACSWQ000000000	[14]
E325-754	JACSWP000000000	[14]
E325-ad1	JACSWO000000000	[14]
E325-ad2	JACSWN000000000	[14]
*Pantoea eucalypti*		
Pv21-6	JACSXJ000000000	[28]
*Erwinia gerundensis*		
Eg13-2	JACSXP000000000	[28]
Eg13-3	JACSXO000000000	[28]
Eg31-2	JACSXG000000000	[28]
Eg31-5	JACSXD000000000	[28]
*Rahnella aquatilis*		
Ra9-2	JACSXQ000000000	[28]
*Pantoea* sp.		
Psp39-30	JACSWU000000000	[28]

**Table 2 microorganisms-08-01449-t002:** Phages used in this study.

Phage	Family	Genus	Species	Accession Number ^a^	Propagation Host	Reference
ϕEa10-1	*Myoviridae*	*Kolesnikvirus*	*Erwinia virus Ea214*	NA	Ea17-1-1	[29]
ϕEa10-2				NA	Ea6-4	[29]
ϕEa21-4				NC_011811.1	Ea6-4	[29,30]
ϕEa45-1B				NA	Ea6-4	[29]
ϕEa35-70		*Agricanvirus*	*Erwinia virus Ea35-70*	NC_023557.1	Pa39-7	[29,31]
ϕEa10-7	*Podoviridae*	*Eracentumvirus*	*Erwinia virus Era103*	NA	Ea29-7	[29]
ϕEa31-3				NA	Ea29-7	[29]
ϕEa46-1-A1				NA	EaD7	[29]
ϕEa46-1-A2				NA	EaD7	[29]
ϕEa9-2		*Johnsonvirus*	*Erwinia virus Ea9-2*	NC_023579.1	Ea17-1-1	[29]

NA: No accession number available, genomes are not deposited. ^a^ The accession number for ϕEra103 is NC_009014.1.

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
