# Peer review of "Population Dynamics between Erwinia amylovora, Pantoea agglomerans and Bacteriophages: Exploiting Synergy and Competition to Improve Phage Cocktail Efficacy"

_microorganisms, 2020, doi:10.3390/microorganisms8091449_

Round 1
Reviewer 1 Report
This is a nicely written manuscript describing developing and characterizing a phage and carrier host strain cocktail for biocontrol of bacterial pathogen Erwinia amylovora. While the study was well-designed and conducted, the data analysis in general lacks statistical testing, therefore the conclusions drawn from the data are not substantiated. Below are my detailed comments.
Please spell out MOI (multiplicity of infection?) where it first appears in the manuscript and provide a brief explanation of it.
Line 61, a reference should be provided to support the statement “Management of fire blight largely consists of alteration of the microbiota resulting in pathogen reduction on the open blossoms”
Line 73, “non-infected cells” needs to explanation. I am wondering if the bacterial host consists of the same genotype, how can some cells not getting infected.
More background information should be included for introducing Erwinia lytic phages, Myoviridae and Podoviridae, such as their infection cycles or factors determining their host ranges.
Table 2, suggest move the column “Accession Number” to the second column position. The current arrangement is not intuitive to understand the Accession Numbers are actually corresponding to several of the Phages, rather than Isolation Hosts.
Section “2.3 Quantitative Real-Time PCR (qPCR)” although the qPCR procedures were likely the same as in reference 24, the primers used in this study should still be reported as a supplemental table along with this manuscript.
Line 167, “MOIg…… was calculated by dividing the genome quantity of phage by that of bacteria.” Since the data were log10 transformed, the calculation is subtraction instead of dividing.
Section “2.7 Data Analysis”, for all the software and packages, please report the version numbers, as this information is critical for reproducibility of the analysis.
Figure 1, I suggest each plot need to be stretched vertically and reduce the thickness of the curves to better visualize individual curve. This issue also applies to Figure 4. The synergistic effect of ɸEa35-70 with ɸEa21-4 and ɸEa46-1-A1 is very interesting. However, statistical testing needs to be carried out to substantiated the conclusion.
Figure 2, again, the bacterial titre values at 24 h need to be statistically compared to support the conclusion.
Line 246 to 247, “…were the only phages which did not increase by at least 2 log units on any hosts tested” Please specify what measurement was performed.
Figure 3, suggest adding labels for bacterial species names for the strains tested, which will better connect the heatmap interpretation to the result description in the main text.
Line 262, “Together, we hoped…” does not sounds appropriate in a conclusion sentence.
Line 269 to 270, I disagree with the statement “Instead of Pa31-4 thriving as Pa39-7 was killed as expected…”. Pa31-4 growth is indeed delayed, but it was likely not killed, because by 24 hrs the bacterial titre of Pa39-7 is visually similar to no phage group (again, no statistical support).
Figure 6, this is an very interesting experiment, but no statistical support renders the conclusion drawn from the results unsubstantiated.
Figure 7, it is very confusing based on the figure caption and main text description regarding what the labels mean. Please specify in the figure caption.
Section “3.5 Comparison of qPCR and Spectrophotometry for Quantification of Mixed Bacterial Populations” this entire section feels disconnected from the rest of the study. What’s the motivation to compare OD vs. qPCR? Also the paragraph ended without a conclusion sentence.
Reviewer 2 Report
The paper presented for review entitled: "Population dynamics between Erwinia amylovora, Pantoea agglomerans and bacteriophages: Exploiting synergy and competition to improve phage cocktail efficacy", concerns the use of bacteriophages as a potential element of E. amylovora control strategy which causes fire blight in apple trees. The study was conducted on 6 isolates of Erwinia amylovora, 30 isolates of E. gerundensis, Rahnella aquatilis, Pantoea sp, Pantoea vagans, Pantoea agglomerans and 10 bacteriophages.
The analyses consisted of: 1. preparation of the material for research, 2. perform qPCR tests, 2 Liquid Culture Growth Experiments, 3. Phage Host Range Assay, 4. Genomic Sequencing of P. agglomerans and Other Epiphytes, 5. Data analysis
The tests carried out by the authors are exclusively laboratory work, which should be carried out in a minimum of 3 series to ensure repeatability of results. In my opinion the work deserves to be published, however, I have a few recommendations and questions to the authors.
Remarks
In my opinion, the methodology of the study needs to be tidied up and refined and linked to the presentation of the results.
L 95-177. Materials and Methods
- qPCR analyses were performed to verify the results of experiments: Liquid Culture Growth Experiments and Phage Host Range Assay therefore a description should be included after the description of these experiments.
- Was the Liquid Culture Growth Experiments and Phage Host Range Assay performed in series? How many series were performed in the experiment?
- Liquid Culture Growth Experiments. What bacteria and strains were used in the study? What phage populations? How many repetitions per test variant? What time periods were the tests performed? This information is missing and therefore the results are not readable.
- Phage Host Range Assay. What was the material for the study? Please provide specific information about Phages and Hosts.
- The qPCR tests are briefly described. Was DNA isolation performed? Please include information about the method used and the number of repetitions per test variant. Standard curves should be included in the materials supplement. In which apparatus were the analyses performed? What was the control for the qPCR reaction? On the basis of the description, it is not known what was tested by the PCR method, although the authors refer to the work of Gayder, et al. 2019, in the manuscript I recommend to include a table with starters which will provide insight into the scope of research work.
- Genomic Sequencing of P. agglomerans and Other Epiphytes. I do not see the results of these analyses in the paper. What was their purpose? Please present the results.On the basis of Table 1, it seems that the bacterial strains used by the authors had full genetic characteristics previously performed.
L. 179-371 Results
The results of the study are presented in four subsections: 3.1. Population Dynamics of the Pathogen E. amylovora and Phage Cocktails, 3.2 Host Range of Phages on Potential Carriers, 3.3 Population Dynamics Between P. agglomerans and Different Phage Cocktails, 3.4 Phage and Carrier Combinations for E. amylovora Reduction, 3.5 Comparison of qPCR and Spectrophotometry for Quantification of Mixed Bacterial Populations.
- The presentation of results does not correspond to the methodology of the experiments.
- There is no information on how Phage Cocktails were prepared in the methodology? Which Phages were combined with which ones?
- The results of the research are interesting, however, the paper does not present all the results of the conducted analyses, which my task should be included in the suplemantery files.
- L. 216-217. Information about calculating the MOI ratio should be included in the methodology. In line 129 there is a brief information without explanation what the MOI ratio is and how it was calculated.
- The authors stated that phage ɸEa35- 70 with both the Myoviridae ɸEa21-4 and Podoviridae ɸEa46-1-A1 and was most effective in combination at reducing E. amylovora growth over 24 h. I have a question for what reason the research was conducted in 24 h only? Have bacteria viability tests been performed?
- Fig. 7. there is no information about PCA analysis in the methodology. The description should be placed in chapter Data analysis including the software used.
- Fig. 8. There is no description of correlation/ regression analysis in the methodology. The description should be placed in chapter Data analysis including the software used in this analysis.
- The authors claim that phage cocktails or incorporating phages can be enabled IPM strategies. Are there any experiments planned on plants? In my opinion, such research should be considered in future research project.
Reviewer 3 Report
The MS “Population dynamics between Erwinia amylovora, Pantoea agglomerans and bacteriophages: Exploiting synergy and competition to improve phage cocktail efficacy” by Gayder et al. represents an interesting contribution to the understanding of how bacteriophages can be used to control plant bacterial pathogens, also exploiting the antagonism shown by some epiphytes against plant pathogens. The experimental model was solid, though there were many variables included in experiments, that is, many strains of the two bacteria and of phages. The high number of variables makes the reading of the MS difficult, though the authors managed to provide a color-coded solution to describe the rather complicated assays. On the other side, if I got it right, all experiments were conducted only once, with 3 replicates/experiment (see below major comment). If this is true, it would have been nice to see that some combinations, the most successful ones, were tested in 3 independent experiments, for example those using the phages ɸEa21-4 and ɸEa46-1-A1. Even if experiments were conducted at different MOIs, for example, MOIs of 10, 1, and 0.01, the experiments should have been repeated to gain more confidence.
Major
Line 147-148 All tests were performed with at least three biological replicates, each measured with qPCR once.
Lines 154-155. All combinations of host and phage were performed with three biological replicates, each measured with qPCR once.
All these experiments were conducted only once, and each combination had three biological replicates? None of these experiments were repeated?
Minor
Line 56 Space between tree and [9]
Line 63 Replace “geopolitical regions” with something like “many countries”
Lines 119-120. One cannot be sure if the citation 24 for the qPCR conditions refers also to the origin of plasmid pTotalStdA. A citation for the plasmid pTotalStdA will add clarity to this issue. Is it also citation 24, Gayder et al 2019 Viruses 11(10):910?
Lines 186-193. The color-coded description of the phages used in experiments should be included in the results section somewhere in lines 184-185 or before, in the material and methods section. It should not appear only in the figure legend is also quite hard.
Lines 351-371. Section 3.5. There is no need to include color-coded description for EaMix and PaMix. There are only two alternatives, easy to follow. The info provided in the legend of Figure 8 is sufficient.
Figure 1, 2, 4, 5, and 6 and Supplemental Figures 1 to 4. “Error bars are the SD around the mean of three replicates”. It is awkward to say “SD around the mean”. SD is the spread of observations around the mean. When indicating SD it is clear that it represents the variation in the dataset from the mean. Please reformulate this sentence in all figure legends.
Round 2
Reviewer 2 Report
The manuscript has been significantly improved. It may be published in its current form. Congratulations to the Authors on well-done research.
Reviewer 3 Report
The authors addressed my suggestions/comments.